Polyphenolic extract from Punica granatum peel causes cytoskeleton-related damage on Giardia lamblia trophozoites in vitro

Palomo-Ligas Lissethe 1
Estrada-Camacho Job 1
Garza-Ontiveros Mariana 1
Vargas-Villanueva José Roberto 1
http://orcid.org/0000-0001-9243-1390 Gutiérrez-Gutiérrez Filiberto 2
Nery-Flores Sendar Daniel 1
Cañas Montoya Jorge Arturo 3
Ascacio-Valdés Juan 1
Campos-Muzquiz Lizeth Guadalupe 1
Rodriguez-Herrera Raul 1 raul.rodriguez@uadec.edu.mx
1 Departamento de Investigación en Alimentos, Facultad de Ciencias Químicas, Universidad Autónoma de Coahuila , Saltillo, Coahuila , Mexico
2 Departamento de Química, Centro Universitario de Ciencias Exactas e Ingenierías, Universidad de Guadalajara , Guadalajara, Jalisco , Mexico
3 Departamento de Polímeros, Facultad de Ciencias Químicas, Universidad Autónoma de Coahuila , Saltillo, Coahuila , Mexico
Braga Erika
Electronic publication date: 2022 Apr 27
Publication date: 2022
Volume: 10
Electronic Location ID: e13350
Received 2022 Jan 21; Accepted 2022 Apr 7
Copyright: © 2022 Palomo-Ligas et al.
Copyright year: 2022
Copyright holder: Palomo-Ligas et al.
License: This is an open access article distributed under the terms of the Creative Commons Attribution License, which permits unrestricted use, distribution, reproduction and adaptation in any medium and for any purpose provided that it is properly attributed. For attribution, the original author(s), title, publication source (PeerJ) and either DOI or URL of the article must be cited.
License URL: https://creativecommons.org/licenses/by/4.0/

Keywords: Giardia lamblia, Antiparasitic, Pomegranate, Cytoskeleton, Tubulin, Ellagic acid, Punicalagin, Microtubule

Funding: FON.SEC.SAGARPA-CONACYT CV-2015-4-266936 CONACYT CV-2021-316124 This study received financial support from The Secretary of Agriculture, Fishing and Livestock Mexico, through the project FON.SEC.SAGARPA-CONACYT CV-2015-4-266936 and CONACYT CV-2021-316124. The funders had no role in study design, data collection and analysis, decision to publish, or preparation of the manuscript.

==============================
Background

Diarrheal diseases caused by protozoa have a great impact on human health around the world. Giardia lamblia is one of the most common flagellates in the intestinal tract. Factors such as adverse effects to first-line drugs or the appearance of drug-resistant strains, make it necessary to identify new treatment alternatives. Agroindustry waste, like pomegranate peel, are a source of phenolic compounds, which possess antiparasitic activities. In vivo studies demonstrated antigiardiasic potential by reducing cyst shedding and protecting intestinal cells; however, they did not identify the compounds or elucidate any mechanism of action in the parasite. The objective of this study is to identify potential molecular targets and to test the in vitro effects of polyphenols from Punica granatum on Giardia lamblia.

Methods

The in vitro antigiardial potential of polyphenolic extract from pomegranate peel (Punica granatum L.) obtained using microwave-ultrasound methodology was evaluated on Giardia lamblia trophozoites. Extract phytochemical identification was performed by HPLC/MS analysis. The effect of polyphenolic extract on growth and adhesion capacity was determined by parasite kinetics; morphological damage was evaluated by SEM, alteration on α-tubulin expression and distribution were analyzed by western blot and immunofluorescence, respectively.

Results

The pomegranate peel extract showed the presence of ellagitannins (punicalin and punicalagin, galloyl-dihexahydroxydiphenoyl-hexoside), flavones (luteolin), and ellagic acid, that caused an inhibitory effect on growth and adhesion capacity, particularly on cells treated with 200 µg/mL, where growth inhibition of 74.36%, trophozoite adherence inhibition of 46.8% and IC50 of 179 µg/mL at 48 h were demonstrated. The most important findings were that the extract alters α-tubulin expression and distribution in Giardia trophozoites in a concentration-independent manner. Also, an increase in α-tubulin expression at 200 µg/mL was observed in western blot and diffuse or incomplete immunolabeling pattern, especially in ventral disk. In addition, the extract caused elongation, disturbance of normal shape, irregularities in the membrane, and flagella abnormalities.

Discussion

The pomegranate peel extract affects Giardia trophozoites in vitro. The damage is related to the cytoskeleton, due to expression and distribution alterations in α-tubulin, particularly in the ventral disk, a primordial structure for adhesion and pathogenesis. Microtubule impairment could explain morphological changes, and inhibition of adhesion capacity and growth. Besides, this is the first report that suggests that ellagic acid, punicalin, punicalagin and luteolin could be interactioning with the rich-tubulin cytoskeleton of Giardia. Further investigations are needed in order to elucidate the mechanisms of action of the isolated compounds and propose a potential drug alternative for the giardiasis treatment.

Introduction

Worldwide, protozoan parasite Giardia lamblia is a common cause of gastrointestinal disease. It proliferates in an extracellular and noninvasive fashion in the small intestine of vertebrate hosts, causing the diarrheal disease known as giardiasis (Cernikova, Faso & Hehl, 2018). Lack of surveillance studies, low sensitivity of diagnostic tools, and resistance to giardiasis treatment add to the challenge in managing giardiasis, leaving a gap that continues to render giardiasis a silent threat to public health worldwide (Roshidi et al., 2021). Despite the existence of diverse antigiardial drugs, most of them present side effects. Within this classification, metronidazole is the most used drug to treat giardiasis. Different studies have shown that metronidazole causes serious side effects on some patients, for example neurotoxicity, optic neuropathy, peripheral neuropathy, and encephalopathy (Hernández Ceruelos et al., 2019). For these reasons, the relevance of finding molecules with potential antiprotozoal/antigiardial activity is increasing with time. Polyphenols are secondary metabolites commonly found in plants that have different biological activities, such as antioxidants, antibacterial, antiparasitic, antiviral and anti-inflammatory effects (Leyva-López et al., 2020). Medicinal plants and agro-industrial wastes such as leaves, roots, tubers, skin, pulp, seeds, and pomace, are an important source of these by-product phytochemicals (Sagar et al., 2018). Agricultural industries generate approximately 76 million tons of waste made up of mainly organic matter, exploitation of agro-industrial waste in search for bioactive compounds promotes green chemistry and furthers the relationship between the agricultural industry and the scientific community (Leyva-López et al., 2020). Pomegranate belongs to the Punicaceae family and is the smallest plant family that includes one genus and two species, including the following: Punica granatum and Punica protopunica (Shaygannia et al., 2016). Pomegranate peel (PP) is a by-product often considered as a waste, comprising nearly 30–40% portion of the fruit (Singh et al., 2018). Studies in vitro and in vivo have shown that total extracts or polyphenolic compounds from pomegranate had cytotoxic effects on protozoa. Ethanolic extracts from pomegranate peel inhibited Plasmodium falciparum and Toxoplasma gondii parasitic growth (Leesombun, Boonmasawai & Nishikawa, 2019). In G. lamblia, in vivo studies showed that alcoholic extracts from P. granatum are effective for the reduction of cysts shedding in giardiasis models in mice and rats and demonstrated the restoration of villi structure, decreased infiltration of lymphocytes, and protection of intestinal cells from apoptotic cell death (Al-Megrin, 2017, El-Kady et al., 2021). However, they did not identify the compounds or elucidate any mechanism of action in the parasite. The objective of this study is to identify potential molecular targets and to test the in vitro effects of polyphenols from P. granatum on G. lamblia.

Experimental Procedures

Plant material and processing

Pomegranates (Punica granatum L.) were commercially acquired in Cuatrociénegas, Coahuila, México. The collected vegetable sample was cleaned and dehydrated in an oven at 50 °C, later the material was grinded until a reduction of the particle size of 0.6 mm.

Phytochemical extraction using microwave/ultrasound

The ethanolic P. granatum extract was obtained using a mass/volume (m/v) relation of 1:12. In this step, 83.33 g of the dry sample was resuspended in 1 L of 30% ethanol-water. The solution was transferred to a 1.5 L crystal reactor and placed in an Ultrasonic Microwave Reaction System (Nanjing ATPIO Instruments Manufacture Co. Ltd., Nanjing, China) under the conditions described before by Valdez-Guerrero and cols (Yathzamiry et al., 2021): Power Radio 20, Ultrasonic on Relay 10, Ultrasonic Off Relay 3, Amplitude Transformer 25, and Set 20 for Ultrasound and Power Radio 800, Display Power 0, Set Time 70 °C and Holding Time 5 for Microwaves. After this treatment, the ethanolic extract was filtered using Whatman No.41 filter paper (Cytiva, Uppsala, Sweden) and stored at 4 °C in amber recipients.

Polyphenolic fraction obtained by Amberlite

The polyphenolic fraction was obtained in a chromatographic column (Merck KGaA, Darmstadt, Germany) where Amberlite XAD-16 resin (MilliporeSigma, Darmstadt, Germany) was used as a stationary phase. The ethanolic P. granatum extract was passed through the column, using distilled water for the removal of carbohydrates, lipids, and other impurities. Later, the phenolic fraction was recuperated from the resin using 96% ethanol as eluent. After that, the ethanol was evaporated from the phenolic fraction using an oven at 45 °C for 12 h until a fine powder was obtained (Hernández-Hernández et al., 2020). The polyphenolic powder was stored in amber bottles at 4 °C to avoid compound degradation.

Identification of polyphenolic compounds by HPLC/MS

The characterization of PP polyphenolic fractions was performed by reversed-phase high-performance liquid chromatography using an HPLC system Varian ProStar (Agilent, Santa Clara, CA, USA) with a diode array detector (280 nm) and mass spectrometer (MS) with a liquid chromatography ion trap Varian 500-MS IT (Agilent, Santa Clara, CA, USA) equipped with an electrospray ion source. The sample (2 mg) was dissolved in 1 mL of 96% ethanol, filtered by 0.45 µm nylon membranes and injected into a Denali C18 column (150 × 2.1 mm, 3 µm, Grace, Albany, OR, USA) maintaining a temperature at 30 °C. The eluents were formic acid (0.2%, v/v) and acetonitrile. The conditions of elution steps were made according to (Ascacio-Valdés et al., 2016). The MS experiments has been made in the negative mode [M-H]-using nitrogen and helium as nebulizing gas and damping gas, respectively. The ion source parameters were spray voltage 5.0 kV and capillary voltage, and temperature were 90.0 V and 350 °C, respectively. The collection and process of the data were done using MS Workstation software Version 6.9 (Agilent, Santa Clara, CA, USA). Sample was analyzed in full scan mode acquired in the m/z range 50–2,000. MS/MS analyses were performed on a series of selected precursor ions. Compounds identification was performed comparing the results with a database of bioactive compounds (WorkStation version 2.0 database, VARIAN, Palo Alto, CA, USA) (Ascacio-Valdés et al., 2016).

Giardia culture and growth inhibition assay

Trophozoites of G. lamblia (ATCC 50803) were axenically grown at 37 °C in borosilicate culture tubes (Thermo Fisher Scientific, Waltham, MA, USA) containing TYI-S-33 medium, pH 7.1, supplemented with 10% bovine serum and 0.5 mg/mL bovine bile (Keister, 1983). Cultures were maintained by subculturing twice a week. In order to determine the effect of polyphenolic extract from P. granatum on G. lamblia trophozoites growth, an inoculum of 350,000 cells from log-phase of growth were incubated for 24, 48 and 72 h at 37 °C with different concentrations of the polyphenolic extract (0, 125, 150, 175, and 200 µg/mL), 0.1% dimethyl sulfoxide (DMSO) (MilliporeSigma, Darmstadt, Germany) and 1.5 µg/mL metronidazole (MTZ) (PiSA, Guadalajara, Mexico) were used as diluent and positive control, respectively. After the incubation times, trophozoites were detached by 30 min of cooling in an ice-water bath and counted by optical microscopy (Carl Zeiss AG, Oberkochen, Germany) using a hemocytometer. Experiments were performed by triplicate and repeated at least twice. Results were expressed as the total number of cells and as the percentage of trophozoites growth in comparison with the control tube (without treatment), also, the 50% inhibitory concentration (IC50) values were calculated by regression analysis.

Adherence inhibition assay

To determine the effect of polyphenolic extract on the capacity of G. lamblia trophozoites to attach into inert superficies, cells treated with P. granatum extract at 0, 125, 150, 175, and 200 µg/mL, and 0.1% dimethyl sulfoxide (DMSO) (MilliporeSigma, Darmstadt, Germany) were grown at 24, 48 and 72 h. After these times, medium with non-adhered trophozoites was removed and maintained on ice for 30 min, after this, tubes were filled with phosphate buffered saline (PBS) and cooled on ice bath for 30 min to detach the adhered cells. The number of trophozoites (adhered and non-adhered) were counted by optical microscopy using a hemocytometer. Results were expressed as percentage of adherence inhibition in relation to the control tube. Experiments were performed by triplicate and repeated at least twice.

The effect of pomegranate peel extract on morphology by Scanning Electron Microscopy (SEM)

Morphologic damage alterations were analyzed using trophozoites incubated for 48 h in 0.1% dimethyl sulfoxide (DMSO) (MilliporeSigma, Darmstadt, Germany) and treated with 150, 175, and 200 µg/mL of the PP extract. Cells were collected, PBS-washed, fixed with 2.5% glutaraldehyde (MilliporeSigma, Darmstadt, Germany) in PBS for 1 h, and adhered to 0.2% poly-(ethylenimine) (MilliporeSigma, Darmstadt, Germany)-coated coverslips. Fixed trophozoites were dehydrated using ethanol increasing concentrations (50–100% v/v) and dried to critical point with CO2. Samples were coated with a thin layer of gold and analyzed by FE-SEM-Hitachi (Hitachi High-Technologies Corporation, Tokyo, Japan).

Effect of pomegranate peel extract on α-tubulin expression

Trophozoites 48 h-treated with 150, 175, and 200 µg/mL of PP extract or 0.1% dimethyl sulfoxide (DMSO) (MilliporeSigma, Darmstadt, Germany) were used to obtain total protein extracts using the NP-40 buffer (MilliporeSigma, Darmstadt, Germany). The protein concentration was determined by BCA Protein Assay Kit (Thermo Fisher Scientific, Waltham, MA, USA), and 10 µg of the protein samples were separated by 10% (w/v) SDS-PAGE using Tris-glycine buffer (Santa Cruz Biotechnology, Dallas, TX, USA) and transferred to PVDF membranes. Protein transfer was performed in PVDF membrane (Millipore, Burlington, VT, USA) using the Trans-Blot® SD Semi-Dry Transfer Cell (Bio-Rad Laboratories, Hercules, CA, USA). Membrane was blocked with 5% BSA/0.1% PBS-Tween (PBST) (FAGA-Lab, Sinaloa, Mexico) solution for 1.5 h at room temperature, washed with PBST 0.1% and incubated with primary antibody (1/500 mouse anti-α-tubulin) (Santa Cruz Biotechnology, Dallas, TX, USA) for 18 h. Membrane was washed again with PBST and incubated for 1 h with 1/5,000 of anti-mouse m-IgGκ BP-HRP (horseradish peroxidase) secondary antibody (Santa Cruz Biotechnology, Dallas, TX, USA). Membrane was washed with PBST and the signal was detected by chemiluminescence using the kit Immobilon ECL Ultra Western HRP Substrate (Millipore, Burlington, VT, USA). Glyceraldehyde-3-phosphate dehydrogenase (GAPDH) antibody (Santa Cruz Biotechnology, Dallas, TX, USA) was used as protein loading control (1/200) (Yang et al., 2002). Experiments were done by triplicate.

The images were captured with the Vision Works version 8.17.16133.9147 Software by Analytik Jena 2016 (www.analytik-jena.de). Semi-quantitative evaluation was performed by densitometry, the GelQuant.NET version 1.8.2 software (BiochemLabSolutions.com, San Francisco, CA, USA) was used.

Effect of pomegranate peel extract on α-tubulin distribution

To analyze the pattern of α-tubulin distribution, trophozoites treated 48 h with 125, 150, 175, and 200 µg/mL of PP extract, DMSO, or albendazole 0.5 µM (MilliporeSigma, Darmstadt, Germany) were analyzed by immunofluorescence (IFA). Samples were collected and counted in a hemocytometer to allow 500,000 cells to adhere in coverslips treated with 0.2% poly-(ethylenimine) (MilliporeSigma, Darmstadt, Germany) for 30 min at 37 °C. Trophozoites were fixed using methanol for 10 min at −20 °C (MilliporeSigma, Darmstadt, Germany). Once dried, the samples were permeabilized in 0.5 Triton X-100 (MilliporeSigma, Darmstadt, Germany) 15 min, and blocked with 1% bovine serum albumin (Santa Cruz Biotechnology, Dallas, TX, USA) for 1 h. Later, samples were incubated 2 h at room temperature with 1/200 mouse anti-α-tubulin IgG1ƙ monoclonal antibody (Santa Cruz Biotechnology, Dallas, TX, USA), washed three times with PBS, and incubated 1 h with 1/200 anti-mouse IgGκ light chain binding protein (m-IgGκ BP) conjugated to fluorescein isothiocyanate (FITC) (Santa Cruz Biotechnology, Dallas, TX, USA), and washed twelve times with PBS. Mounting of the preparations were done using Prolong Gold with DAPI (Thermo Fisher Scientific, Waltham, MA, USA). Preparations were analyzed by fluorescence Axiolab A1 35 LED Binocular Microscope (Carl Zeiss™, Oberkochen, Germany) and image acquisition was done using the software, ZEN 2.3 edition.

Statistical analysis

The data from growth, adherence inhibition and western blot were analyzed by GraphPad Prism 6 (GraphPad Software, San Diego, CA, USA) using the two-way ANOVA test and as post hoc the Tukey test, statistical significance was considered with P values ≤ 0.05.

Results

Identification of polyphenolic compounds in pomegranate peel extract by HPLC/MS

Phytochemical isolation was performed by microwave-ultrasound hybridization using ethanol as a solvent. The obtention of enriched polyphenolic extract was realized by liquid chromatography with Amberlite XAD-16. The chromatograms obtained through HPLC system and mass spectrometer were compared on a database of bioactive compounds. The characterization of pomegranate peel showed the presence of ellagitannins (punicalin and punicalagin isomers, galloyl-dihexahydroxydiphenoyl-hexoside), flavones (luteolin 6-C-glucoside), and ellagic acid (including ellagic acid-hexoside) (Table 1).

Table 1 HPLC/MS analysis from polyphenolic extract of pomegranate peel.

Sample	RT	Mass (m/z)	Compound	Family	
POMEGRANATE PEEL	4.809	781	Punicalin	Ellagitannins	
	12.466	780.9	Punicalin	Ellagitannins	
	15.322	1,083	Punicalagin	Ellagitannins	
	16.119	1,083	Punicalagin	Ellagitannins	
	21.019	633	Galloyl-dihexahydroxydiphenoyl-hexoside	Ellagitannins	
	22.469	463	Ellagic acid-hexoside	Ellagitannins	
	26.981	447	Luteolin 6-C-glucoside	Flavones	
	28.048	300.9	Ellagic acid	Hydroxybenzoic acid dimers	

Polyphenolic extract effects on Giardia lamblia trophozoite growth

Results showed that the tested extract inhibited the parasite growth, all the extract concentrations used in this study were statistically different in comparison to non-treated cells at the times tested. The effect was evident at 24 h, but at 48 h, it was observed the maximal effect in comparison with negative control. There was no difference between non-treated cells and cells grown with the diluent of the extract. Trophozoites treated with 200 µg/mL of pomegranate peel extract showed an effect like MTZ at 48 and 72 h (69.09% and 68.15% of inhibition, respectively), showing significant growth inhibition of 74.36% at 48 h and 66.32 at 72 h (Fig. 1). The calculated IC50 of the extract was 179 µg/mL at 48 h. These results demonstrate that the polyphenolic extract from the studied vegetable source, has an in vitro inhibitory effect on G. lamblia growth.

Figure 1 Effect on G. lamblia trophozoites growth by pomegranate peel extract.

Data analyzed with GraphPad 6 software, *p ≤ 0.001, **p ≤ 0.0001.

Polyphenolic extract affects trophozoite adherence to inert surfaces

To establish if the studied extract influences trophozoite adherence to inert surfaces, growth kinetics were performed, treating trophozoites with the polyphenolic extract. Results showed that the extract reduced adherence since 24 h, except for 125 µg/mL, in comparison with cells without treatment. The higher effect on adherence inhibition was observed at 48 h in a dose-dependent manner, where the 200 µg/mL treatment was able to inhibit G. lamblia trophozoite adherence in 46.8% (Fig. 2). These results demonstrate that the tested polyphenolic extract was able to in vitro inhibit G. lamblia trophozoite adherence.

Figure 2 Effect on G. lamblia trophozoites adhesion capacity by pomegranate peel extract.

The results were expressed as the percentage of adherence inhibition in relation to the control. Data analyzed with GraphPad 6 software, *p ≤ 0.01, **p ≤ 0.001, ***p ≤ 0.0001.

Pomegranate peel extract alters the morphology on Giardia lamblia trophozoites

To evaluate alterations in G. lamblia trophozoite morphology caused by the extract; cells were treated with polyphenolic pomegranate peel extract and DMSO as negative control for 48 h, then, they were visualized using SEM. Parasites grown in solvent showed normal pear shape (Fig. 3A). In treated cells, an elongation, loss of shape, and protrusions on the dorsal surface were observed. Trophozoites incubated with the extract had irregularities on the periphery and showed vesicle-like structures on the dorsal surface. In addition, abnormalities on the caudal region and flagella deformations, including shortening or loss, compared with control cells were detected (Figs. 3B–3D). Pomegranate extract also appears to cause holes on the membrane (perforation), this effect increased with concentration (Fig. 3D).

Figure 3 SEM micrographs of morphological alterations in G. lamblia trophozoites by pomegranate peel extract.

(A) DMSO, (B) 150 µg/mL, (C) 175 µg/mL, and (D) 200 µg/mL treated cells. PP extract caused elongation, loss of normal shape, edge irregularities, membrane perforations (arrowhead) and abnormal length (asterisk) or loss of flagella.

Tubulin G. lamblia alteration by pomegranate peel extract

In order to determine changes in cytoskeleton protein expression, α-tubulin was detected by Western Blot. It was found an alteration on α-tubulin expression levels by pomegranate extract treatment in trophozoites with 150 and 200 µg/mL in comparison with DMSO incubated cells. Interestingly, it is observed that treatments affect α-tubulin expression in a concentration-independent manner (Fig. 4), where the lowest concentration decreased protein expression, but the highest concentration increased it. At 175 µg/mL no differences were found. Another additional finding is shown in Fig. S1 where glyceraldehyde-3-phosphate dehydrogenase (GAPDH) was attempted to be used as a loading control, however, the samples treated with the PP extract increased the expression of this protein.

Figure 4 Alteration of α-tubulin expression in G. lamblia trophozoites by pomegranate peel extract.

Image of western blotting (shown above) and densitometric analysis of (A) DMSO, (B) 150 µg/mL, (C) 175 µg/mL, and (D) 200 µg/mL treated cells. Data analyzed with GraphPad 6 software, *p ≤ 0.0001.

Tubulin distribution of G. lamblia was affected by pomegranate peel extract

Immunofluorescence assay was performed in order to analyze alterations on the cell localization of α-tubulin on PP treated trophozoites. In non-treated cells (Fig. 5A–A2), α-tubulin is located on flagella, median body and ventral disc (VD), in addition, cells present normal size and shape. Conversely, cells treated with the extract showed differences in the staining pattern, which indicates modifications in G. lamblia microtubule cytoskeleton. Interestingly, these alterations are independent of the concentration, like WB observations. Immuno-labeling of cells treated with PP extract showed interesting events on VD distribution. In 125 μg/mL treated cells with the ventral side exposed, the α-tubulin signal in VD is not localized, nor the bare area is observed (Fig. 5–B2). At 175 µg/mL and 200 µg/mL the intensity of tubulin is lower than control, VD appears incomplete (Fig. 5–D2, asterisk), with diffuse outline, and incomplete cells consisting only of the disc (Fig. 5–E2), were observed. These changes are associated with morphological alterations, where a loss of characteristic pear shape and VD deformation was observed (Fig. 5B–E). In accordance with SEM, flagellar disturbance as bifurcation (Fig. 5–C2, arrow) or structure loss (Fig. 5–E2) were also detected. For ABZ exposure, trophozoites appear completely deformated and increased in size (Fig. 5F-F2).

Figure 5 Effects on α-tubulin distribution in G. lamblia trophozoites by pomegranate peel extract.

(A) DMSO, (B) 125 µg/mL, (C) 150 µg/mL, (D) 175 µg/mL, (E) 200 µg/mL treated cells, and (F) Albendazole. The number indicates (1) DAPI nuclei staining and (2) mouse anti α-tubulin antibody followed by anti-mouse IgGκ light chain binding protein (m-IgGκ BP) conjugated to fluorescein isothiocyanate (FITC). Cells were examinated with a fluorescent microscope under magnification of 400×. Arrowhead indicates ventral disk; asterisks, median body; and arrows, flagella.

Discussion

The cytoskeleton of Giardia is a unique and complex structure essential for life cycle. It is formed by specialized microtubular structures such as the ventral disk, four pairs of flagella, the median body, and the funis (Gadelha, Benchimol & Souza, 2017). Cytoskeleton is involved in motility, adhesion, cell division, encystation, cell shape, cell polarization, and intracellular trafficking (Hagen et al., 2020). Microtubules (MT), are cylindrical structures composed of two proteins of similar mass, α and β tubulin (Gadelha, Benchimol & de Souza, 2020). They are highly dynamic structures that cycle between transition periods of growth and depolymerization (Roll-Mecak, 2020). This dynamic instability is mediated by GTP. MT self-assemble in the presence of GTP and require GTP to be bound to β-tubulin for individual dimers to be added. During polymerization new GTP-αβ-tubulin dimers are added to the plus end of MT, thus forming a GTP-tubulin “cap” at the plus end and stabilizing the MT. If the GTP cap is lost, MT enter a rapid state of depolymerization from the plus end (Kent & Lele, 2017). Due to their importance, alterations in proteins or cytoskeletal elements in the parasite can lead to cellular damage, alteration of attachment, and ultimately cell death. Several phytochemicals have demonstrated their antiparasitic effects through alteration of cellular elements essential for this pathogen survival (Ray & Sarkar, 2017; Wink, 2012). In our study, we analyzed the effect of polyphenols in pomegranate peel ethanolic extract on G. lamblia trophozoites in vitro. Although the antigiardiasic activity of methanolic (Al-Megrin, 2017) and ethanolic extracts (El-Kady et al., 2021) from P. granatum has been reported in vivo, these reports focused on the host response, and they do not identify potential molecular targets in the parasite nor describe molecules responsible for the activity. In our results, the identified polyphenols in pomegranate peel extract tested are the following: ellagitanins (punicalin and punicalagin, galloyl-dihexahydroxydiphenoyl-hexoside), flavones (luteolin), and ellagic acid (Table 1), which are in agreement with other research (Ahmadiankia, 2019). The PP extract inhibits trophozoite growth by 74.36% at a concentration of 200 μg/mL, with an IC50 of 179 μg/mL (48 h) (Fig. 1), and causes morphological alterations, generating elongation in the cell, loss of shape, flagella disturbance, and dorsal protuberances (Fig. 3). Similar alterations have been previously described in the intestinal mucosa of Giardia infected rats, where loss of flagella and distortion in the ventral disk of trophozoites were shown (El-Kady et al., 2021). Due to MT cytoskeleton importance in the parasite, we evaluated the expression, distribution of α-tubulin, and adhesion capacity in Giardia trophozoites. In WB and IFA, the effects of PP extract were independent of concentration. Blots of parasites treated with 200 µg/mL showed an increase while 150 µg/mL showed decrease in α-tubulin expression in comparison to non-treated cells (Fig. 4). Meanwhile, IFA results showed that PP extract causes alterations in the intensity and pattern of α-tubulin staining. We highlight the following alterations: incomplete (Fig. 5–D2) or diffuse immunolabeling on VD (Fig. 5–E2); loosed bare area (in which the entire VD shows immunolabeling), and bifurcated flagella (Fig. 5–C2). The modifications on tubulin distribution in VD of treated cells, can be related with morphological damage where this structure looks specially misshapen (Figs. 5B–5E), and with the inhibition of trophozoite adherence of 46.8% at 48 h (Fig. 2). In addition, α-tubulin overexpression in 200 µg/mL treated cells, observed in WB, could be associated with the elongation of the cell. Our hypothesis is that the increase in cell length or the disorganization in parasite MT structures could be reflected in this increased expression. However, this does not fully explain the concentration-independent effects. In this sense, the identified molecules in the PP ethanolic extract could be related. To our knowledge, punicalagin, luteolin and ellagic acid have reports of their effect on cytoskeleton in other cells, however only the last is related with MT alterations. For example, punicalagin prevents F-actin remodeling and inhibits the migration of fibroblast-like synoviocytes isolated from rheumatoid arthritis patients (Huang et al., 2021). Other investigations prove the effect of luteolin, ellagic acid, and punicic acid combination limiting the invasion of cancer cells. These treatment downregulated fascin, anillin and nexillin, molecules involved with actin cytoskeleton (Rocha et al., 2012; Wang et al., 2012). In the case of microtubules, there are multiple binding sites reported for different drugs, in this way, the interaction at each site generates a different physiological effect. Among the molecules that affect MT, are inhibitors of tubulin polymerization (colchicine, vinblastine) or inhibitors of depolymerization (paclitaxel) (Wink, 2012). The interaction between polyphenols and MT structures has been documented in Giardia with both effects. On the one hand, reduction of α-tubulin expression level in Giardia has been documented with curcumin, causing changes in its distribution in vitro and interactioning close to the vinblastine binding site in silico (Gutiérrez-Gutiérrez et al., 2017a). The same occurs with podophyllotoxin-type lignans from Bursera fagaroides var. fagaroides, which affect the distribution and staining pattern of microtubular structures on Giardia trophozoites, by the union in colchicine binding site (Gutiérrez-Gutiérrez et al., 2017b; Gutiérrez-Gutiérrez et al., 2019). On the other hand, it has been shown that paclitaxel is capable of producing microtubule stabilization, thereby generating a significant increase in the size of both the median body and the eight axonemes (Dawson et al., 2007). In this way, ellagic acid has the ability to increase the amount of polymerized tubulin. This results in an enhanced MT assembly in vitro, in prostate cancer cells (Eskra, Schlicht & Bosland, 2019). Besides, it is also possible that polyphenols affect cytoskeleton regulatory elements. For example, Nek8445 belongs to the large family of Giardia Nek kinases which regulates MT organization, its deletion causes abnormal disk organization, cell shape impairment, short flagella, and lack of funis (Hennessey et al., 2020). Furthermore, an interesting association that can explain our effects is the dual behavior of polyphenols at diverse cellular contexts. The flavonoid epigallocatechin 3-gallate isolated from green tea, stimulates cellular events as intracellular signaling at lower concentrations. By contrast, high concentrations of this compound may cause severe stress which compromises cellular integrity and disrupts nuclear and mitochondrial function (Kim, Quon & Kim, 2014). In addition to the dual effect, it has been described that polyphenols can interact with each other. For example, anthocyanins and tannins compete for the same binding site on plant cell walls. In addition, these unions are also dependent on the exposure time, pH and presence of proteins (Siemińska-Kuczer, Szymańska-Chargot & Zdunek, 2022). Therefore, the concentration-independent effects observed in our study could be altered by competition between polyphenols or the experimental conditions (pH of the medium, presence of bovine fetal serum, among others).

Due to their pleiotropic effects, the polyphenols found in pomegranate extract could have different mechanisms of action. Another theory that explains our results is a stress response or interaction with membrane lipids caused by the presence of polyphenols. The irregularities on trophozoite surface detected by SEM (Fig. 3) could be microvesicles (MV) which are a type of extracellular vesicle originating from a budding of the plasma membrane (de Souza & Barrias, 2020). MVs are often associated with different functions such as cell signaling or as a response to stress stimuli. For example, a study demonstrated an increase in MV production on G. lamblia’s trophozoites exposed to extreme pH levels for the parasite, suggesting that MV are released as a response to cellular stress/damage (Deolindo, Evans-Osses & Ramirez, 2013). A stress related finding suggests that polyphenols from the PP extract caused an increase in GAPDH expression (Fig. S1). It has been described that the polyphenols epicatechin and kaempferol exert their anti-parasitic effect by dysregulation of enzymes involved in energy metabolism in Entamoeba histolytica (Martínez-Castillo et al., 2018). Also, in Leishmania, the mRNA alteration of GAPDH and superoxide dismutase (SOD) appears to be related with oxidative stress. This parasite needs linoleate as precursor to synthetize essential fatty acids, when the NAD(P)H cytochrome b5 oxidoreductase (Ncb5or) enzyme is lacked, the deficiency of linoleate causes increasing H2O2 (accomplished by overexpression of GAPDH and SOD), and membrane depolarization, which is followed by apoptosis and cell death (Mukherjee et al., 2012). As mentioned, the effects caused by polyphenols can be indirect as in the previous study, or directly affect their molecular targets. For example, punicalagin decreases yeast plasma membrane ergosterol causing severe ultrastructural changes on Cryptococcus gattii and Candida albicans (Silva et al., 2020). Otherwise, ellagic acid inhibits sialidase of Trypanosoma congolense (Aminu et al., 2017) and neuraminidase on influenza A virus (Li et al., 2021). Reports about Giardia indicate that oseltamivir, a viral neuraminidase (sialidase) inhibitor, affects GM1 ganglioside, a major constituent of lipid rafts, and disassembled these microdomains (De Chatterjee et al., 2015). Based on these reports, it could be possible that some of the polyphenols in PP extract interfere with lipid rafts in Giardia causing the observed irregularities and perforation on trophozoite membrane (Fig. 3).

In our study, the PP extract alters Giardia trophozoites primordially through cytoskeleton by α-tubulin modification, especially in VD, that causes morphology alterations and compromised adhesion, and therefore parasites survival. This is the first report that associates that ellagic acid, punicalin, punicalagin and luteolin could be interactioning with the rich-tubulin cytoskeleton of Giardia. However, it could be that polyphenols have different effects, inducing a stress response or interacting with membrane microdomains in the protozoa. Further investigations are needed in order to elucidate the mechanisms of action of the compounds. This information is important to identify new possible molecules for giardiasis treatment which can be found in natural sources like the pomegranate peel.

Conclusions

The MT in G. lamblia’s cytoskeleton play an imperative role for the parasite. Alterations of proteins like tubulin directly affect the MT structure and interfere with essential processes for Giardia’s survival. Our results demonstrate that pomegranate peel extract interacts with cytoskeleton by Giardia tubulin alteration. As a result, we observed disturbances in MT structures like ventral disk and flagella, elongation and deformation cells, reduction in the adhesion capacity, and growth inhibition. This study relates for the first time the effect of ellagic acid, punicalin, punicalagin and luteolin with the alteration in MT dynamics by tubulin interaction. However, polyphenols could be interacting with other targets inducing a parasite response to stress, like the vesicle-like structures observed. Further analysis to identify the mechanisms of action caused by these molecules need to be explored. Results from the present study, suggest a list of new molecules that can be tested as potential antigiardiasic agents and, at the same time, promote green chemistry by taking advantage of agro-industry waste to give them another use.

Supplemental Information

Supplemental Information 1 Raw Data.

Click here for additional data file.

Supplemental Information 2 Alteration of GAPDH expression in Giardia lamblia trophozoites by pomegranate peel extract.

Image of western blotting (shown above) and densitometric analysis of (A) DMSO, (B) 150 µg/mL, (C) 175 µg/mL, and (D) 200 µg/mL treated cells. Data analyzed with GraphPad 6 software, *p ≤ 0.0001

Click here for additional data file.

Additional Information and Declarations

Competing Interests

Author Contributions

Data Availability

The authors declare that they have no competing interests.

Lissethe Palomo-Ligas conceived and designed the experiments, performed the experiments, analyzed the data, prepared figures and/or tables, authored or reviewed drafts of the paper, and approved the final draft.

Job Estrada-Camacho performed the experiments, analyzed the data, prepared figures and/or tables, authored or reviewed drafts of the paper, and approved the final draft.

Mariana Garza-Ontiveros performed the experiments, analyzed the data, prepared figures and/or tables, authored or reviewed drafts of the paper, and approved the final draft.

José Roberto Vargas-Villanueva performed the experiments, analyzed the data, prepared figures and/or tables, authored or reviewed drafts of the paper, and approved the final draft.

Filiberto Gutiérrez-Gutiérrez conceived and designed the experiments, analyzed the data, authored or reviewed drafts of the paper, and approved the final draft.

Sendar Daniel Nery-Flores analyzed the data, authored or reviewed drafts of the paper, and approved the final draft.

Jorge Arturo Cañas Montoya performed the experiments, authored or reviewed drafts of the paper, and approved the final draft.

Juan Ascacio-Valdés performed the experiments, analyzed the data, prepared figures and/or tables, authored or reviewed drafts of the paper, and approved the final draft.

Lizeth Guadalupe Campos-Muzquiz analyzed the data, authored or reviewed drafts of the paper, and approved the final draft.

Raul Rodriguez-Herrera analyzed the data, authored or reviewed drafts of the paper, and approved the final draft.

The following information was supplied regarding data availability:

The raw data (full-length images of western blot, kinetics and densitometric analysis) are available in the Supplemental File.

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
