# Peer review of "Polyphenolic extract from Punica granatum peel causes cytoskeleton-related damage on Giardia lamblia trophozoites in vitro"

_PeerJ, doi:10.7717/peerj.13350_

## Round 0.1 · original submission · Major Revisions

The review process is now complete, and three thorough reviews from highly qualified referees are included at the bottom of this letter. Although there is considerable merit in your paper, we also identified some concerns that must be considered in your resubmission. I strongly agree with them and emphasize that the authors must dedicate themselves to answering the points raised with the utmost precision and making all those reconsiderations in the manuscript to be submitted.

Reviewer 1 ·

Basic reporting

1) The summary is unclear:
a) It remains to be argued why pomegranate peel extract is a good option against Giardia trophozoites. Why should it have activity against Giardia, if previously it has only been shown to have antibacterial activity?
b) The results do not describe the compounds identified by HPLC.
c) It is necessary to improve the discussion of the results.

2) The quality of figure 4 is low, the resolution needs to be improved.
3) What is the most relevant contribution, if in the introduction section they mention that the activity of pomegranate peel extract against Giardia was previously demonstrated in an in vivo model?
4) With the above, I believe that there should be a rethinking of the hypothesis.

Experimental design

In relation to the experimental design, I consider that it is well planned for the proposed objectives, I would only observe the following:

1) in section 2.6 Adherence inhibition assay: what concentrations of the extract were used?
2) in section 2.7 The Effect of pomegranate peel extract on morphology by Scanning Electron Microscopy (SEM): The SEM methodology needs to be further detailed.

Validity of the findings

The proposed hypothesis lacks solidity based on the referred antecedents.
The discussion is weak, mainly in the integration of the results, for example, why does the increase in the expression of tubulin destabilize the cytoskeleton? Why are the polyphenolic compounds identified decisive in the decrease in adherence or growth of Giardia?
A cell viability assay after treatment and the evaluation of other molecular markers of cell viability would help to reach more decisive conclusions.

Reviewer 2 ·

Basic reporting

NC

Experimental design

Dear authors,

The overall methodology is logic and acceptable. However, there are comments (all are in yellow highlighted colors) in the attached file for further improvement of this part.

Validity of the findings

Regarding Results, Discussion and References, comments are in yellow highlighted colors for further improvement.

Annotated reviews are not available for download in order to protect the identity of reviewers who chose to remain anonymous.

Reviewer 3 ·

Basic reporting

It is an interesting and well written and structured study; the introduction presents relevant information related to the need for new giardicidal agents and the importance of phenolic compounds as antiparasitic agents. Previous studies (Al-Megrin, 2017; El-Kady et al., 2021), using a methanolic and ethanolic extract from P. granatum, demonstrated the reduction of G. lamblia cysts and trophozoites using a giardiasis model as wells as the induction of nitric oxide (NO). However, the characterization of the possible therapeutic targets of the promegranate extract has not been determined. In the present study, the in vitro giardicidal activity of an ethanolic extract obtained from pomegranate peel was evaluated. Authors identified the polyphenolic compounds contained in the ethanolic extract ( ellagic acid and punicallagins, among other components) and their inhibitory effect on parasite growth and adhesion capacity, as well as its effect on the morphology and tubulin expression of parasite are presented. Figures are relevant, and figure legends are well described.

Experimental design

There are some considerations to the present study.
Different concentrations of the PP extract were tested, it is shown that parasite growth was affected by all the concentrations, as well as the adhesion capacity with the exception of 125ug/mL. Further assays were performed from 150 ug/mL to 200 ug/mL, demonstrating that the extract induced several morphological changes such as holes on the membrane and shorter flagella, among others. In addition, they showed that at 150 and 200 ug/mL, the ethanolic extract has different effect on the alpha tubulin expression demonstrated by Western blot analysis.
- It is convenient that authors state the reason they had to exclude the lowest concentration for ultrastructural and tubulin expression assays. Although the lowest concentration did not affect parasite adhesion, it inhibited the parasite growth, then, it could be of interest to evaluate if the lowest concentration induced any effect on alpha tubulin expression (decreased or overexpressed) or at the ultrastructural level. Besides, authors mentioned that “the structural and functional affectations caused by pomegranate may be due to the change in the expression of tubulin”. In this regard, it is confused to correlate SEM and Western blot analysis; since the three concentrations evaluated (150, 175 and 200 ug/mL) affected the parasite morphology in a similar way, however, their effect on alpha tubulin expression is completely different.. One important feature that requires to be mentioned is how the alpha tubulin expression at different concentrations was normalized? Besides, figure 4 quality needs to be improved.
It could be interesting to include parasites treated with commercial ellagic acid and analyze the effects on parasite at the ultrastructural level and alpha tubulin expression, supporting the suggestion that the ellagic acid of the ethanolic extract induces alterations on the parasite cytoskeleton.

Validity of the findings

Authors mentioned that “These reports lead us to suppose that polyphenols, possible ellagic acid, in PP extract alters Giardia cytoskeleton”. However, data presented herein do not give enough evidence that the extract or the ellagic acid alter the parasite cytoskeleton. No strong evidence supports the effect on parasite microtubule organization (disk alteration, lack of funnis, among others).
In addition, authors discussed the fact that 150 µg/mL and 200 µg/mL exposure caused opposite effects in α-tubulin expression. They mentioned that “paclitaxel-resistant human lung cancer cell lines overexpress α- and β-tubulin” and on the other hand “reduction of α- tubulin expression level in Giardia has been documented with curcumin, another polyphenolic compound, causing changes on its distribution”.
-In this regard, how can authors explain that both concentrations, together with 175ug/mL (that in Western blot analysis do not affect alpha tubulin expression), induce similar morphological alterations?

Additional comments

In general terms, the present study shows data about the effect of ethanolic extract on the growth and adhesion of Giardia trophozoites; in addition, the mechanism of action is analyzed by SEM and Western blot assays however, data presented are preliminary and do not clearly demonstrate the effect on the parasite cytoskeleton, a fact that authors suggest.
To give more experimental support to their suggestion about the effect of the extract on the tubulin network, it is important to demonstrate the effect on the ventral disc as well as other cytoskeleton components. SEM and immunofluorescence studies could give this evidence; SEM assays showing the ventral side of parasite could give relevant information

---

## Round 0.2 · accepted · Accept

The authors have satisfactorily responded to all questions and made the necessary changes to the manuscript.